# Field-Effect Sensors Using Biomaterials for Chemical Sensing

**DOI:** 10.3390/s21237874

**Published:** 2021-11-26

**Authors:** Chunsheng Wu, Ping Zhu, Yage Liu, Liping Du, Ping Wang

**Affiliations:** 1Institute of Medical Engineering, Department of Biophysics, School of Basic Medical Science, Health Science Center, Xi’an Jiaotong University, Xi’an 710061, China; wuchunsheng@xjtu.edu.cn (C.W.); jewel121@stu.xjtu.edu.cn (P.Z.); yageliu@xjtu.edu.cn (Y.L.); duliping@xjtu.edu.cn (L.D.); 2Biosensor National Special Laboratory, Department of Biomedical Engineering, Zhejiang University, Hangzhou 310027, China; 3Key Laboratory for Biomedical Engineering of Ministry of Education, Department of Biomedical Engineering, Zhejiang University, Hangzhou 310027, China

**Keywords:** field-effect sensors, chemical sensors, biosensors, olfactory, taste, biomaterials

## Abstract

After millions of years of evolution, biological chemical sensing systems (i.e., olfactory and taste systems) have become very powerful natural systems which show extreme high performances in detecting and discriminating various chemical substances. Creating field-effect sensors using biomaterials that are able to detect specific target chemical substances with high sensitivity would have broad applications in many areas, ranging from biomedicine and environments to the food industry, but this has proved extremely challenging. Over decades of intense research, field-effect sensors using biomaterials for chemical sensing have achieved significant progress and have shown promising prospects and potential applications. This review will summarize the most recent advances in the development of field-effect sensors using biomaterials for chemical sensing with an emphasis on those using functional biomaterials as sensing elements such as olfactory and taste cells and receptors. Firstly, unique principles and approaches for the development of these field-effect sensors using biomaterials will be introduced. Then, the major types of field-effect sensors using biomaterials will be presented, which includes field-effect transistor (FET), light-addressable potentiometric sensor (LAPS), and capacitive electrolyte–insulator–semiconductor (EIS) sensors. Finally, the current limitations, main challenges and future trends of field-effect sensors using biomaterials for chemical sensing will be proposed and discussed.

## 1. Introduction

Biological olfactory and taste systems are two main categories of natural chemical sensing systems, which play crucial roles for almost all the creatures in survival, feeding, and breeding [1,2,3,4,5]. After millions of years of evolution, these biological chemical sensing systems have become very powerful natural systems which show extreme high performances in detecting and discriminating various chemical substances [2,6,7,8]. For instance, biological olfactory systems are able to detect specific chemical signals presented by the odorant molecules, even at the trace level [9,10]. Similarly, biological taste systems show unique performance and versatility for the detection of chemical signals transmitted by various tastants [2,11]. Creatures are able to obtain essential chemical information about their surroundings from biological chemical sensing systems in order to find food, to communicate with partners, and to avoid predators [12,13,14,15]. The key components of biological chemical sensing systems include functional biomaterials that are able to recognize specific chemical substances and transduce the sensed chemical signals into cellular and molecular responses [2,6,7]. These functional biomaterials, which are chemical sensitive cells and molecules, mainly include olfactory sensory neurons, olfactory receptors, taste cells, and taste receptors [16,17]. They have been considered the primary source of high performances of biological chemical sensing systems [7,18,19].

Creating field-effect sensors using biomaterials that are able to detect specific target chemical substances with high sensitivity would have broad applications in many areas, ranging from biomedicine and environments to the food industry, but this has proved extremely challenging [20,21,22]. The excellent performances of functional biomaterials from biological chemical sensing systems are ideal candidates of sensitive elements for the development of field-effect sensors using biomaterials towards chemical sensing in complex environments [23,24]. For this reason, these biomaterials have been employed for chemical sensing to mimic the mechanisms of biological chemical sensing systems. In recent decades, with rapid advancements in molecular biology and microfabrication process, inspirations from natural chemical sensing systems have led to the development of various field-effect sensors using biomaterials that rely on the combination of functional biomaterials with various field-effect devices [25,26,27,28]. Over decades of intense research, field-effect sensors using biomaterials for chemical sensing have achieved significant progress and shown promising prospects and potential applications.

The development of chemical sensors has been inspired by utilizing the biological sensitive materials or mimicking natural porous structures [29,30,31,32,33]. For the latter situation, many chemical sensors were developed to improve the sensing performance [34]. A typical example is the architecture hierarchy of butterfly wings, which can be synthesized chemically via specific approaches. For example, well-organized porous hierarchical SnO_2_ was fabricated with connective hollow interiors and thin mesoporous walls for the sensing of chemical vapors [35,36]; the photonic structures from Morpho butterfly wings were prepared for the sensitive optical sensing of ethanol [37,38,39,40]. In addition, other biological templates have also been mimicked to fabricate sensitive materials with hierarchical micro/nanostructures, such as the eggshell membrane [41] and the bristles on the wings of the Alpine Black Swallowtail butterfly (*Papilio maackii*) [42]. Considering that the assembly of biological micro/nanostructures mainly belong to the category of material chemistry and has been summarized in other reviews [29,30,43], here we would like to focus on how to utilize the biological sensitive materials with secondary transducers for chemical sensing. Among various chemical sensors, field-effect sensors using biomaterials could retain the biological chemical sensing mechanisms to some extent and could achieve a performance comparable to biological chemical sensing systems by the using of functional biomaterials as sensitive elements for chemical sensing, which are characterized with high sensitivity, high specificity, and low detection limit [16,44].

Despite the rapid advancements and growing interests in the research and development of field-effect sensors using biomaterials for chemical sensing, limited literature is available that outlines recent advances in this field. This review will summarize the state of the art in field-effect sensors using biomaterials for chemical sensing with an emphasis on those using functional biomaterials as sensing elements, such as olfactory and taste cells and receptors. Firstly, unique principles and approaches for the development of these field-effect sensors using biomaterials will be introduced. Then, the major types of field-effect sensors using biomaterials will be presented, which includes field-effect transistor (FET), light-addressable potentiometric sensor (LAPS), and capacitive electrolyte–insulator–semiconductor (EIS) sensors. Finally, the current limitations, main challenges and future trends of field-effect sensors using biomaterials for chemical sensing will be proposed and discussed.

## 2. Fundamental of Field-Effect Sensors Using Biomaterials

In biological chemical sensing systems, the process of chemical signal detection is initialized by the special interactions between molecular detectors and specific chemical substances, which can trigger a cascade of intracellular biochemical reactions to convert the chemical signals into cellular responses such as cell membrane potential changes [45,46,47]. These cellular responses are transmitted to the central neural system for the further processing of chemical signals, which allows for the perception of specific chemical substances. Biological chemical sensing systems are the most powerful system for the detection of specific chemical substances with very high performances that cannot be matched by most existing artificial devices. Therefore, it is worthwhile to develop biosensors using biomaterials in order to obtain artificial chemical sensing devices with performances comparable to biological chemical sensing systems.

The main components of biosensors using biomaterials for chemical sensing include sensitive elements and transducers, which are combined to mimic the functions of biological chemical sensing systems to realize the conversion of chemical signals into measurable signals by existing devices such as electrical signals and optical signals. As shown in Figure 1, the basic idea of biosensors using biomaterials is to employ the extreme high capability of functional biomaterials originating from biological systems for the detection of specific chemical substances. The coupling of highly specialized biomaterials with a transducer could lead to the generation of potential devices and instruments with a performance comparable to that of biological chemical sensing systems for the detection of chemical signals in a trace level within complex environmental conditions.

### 2.1. Preparation of Functional Biomaterials

For the development of biosensors using biomaterials, it is required to obtain functional biomaterials, which maintain their unique capability of chemical sensing and are suitable to be used as sensitive elements to couple with transducers [16]. Because the activity of biomaterials has a direct influence on the performances of biosensors with regard to sensitivity, specificity, and stability, it is of great importance to obtain functional biomaterials for chemical sensing. In addition to maintaining the natural structures and native functions of biomaterials, it is also desirable to produce them in a cost-effective manner and store them in a convenient manner. At present, several methods have been applied in the preparation of functional biomaterials for chemical sensing, which can be divided into two main categories: one is direct isolation from natural biological chemical sensing systems, the other one is preparation based on biotechnology.

Direct isolation from natural biological chemical sensing systems is the most convenient approach to achieving functional biomaterials for the development of biosensors for chemical sensing. It is widely used in the early stage of biosensors, which has the advantages of maintaining the natural structure and functions of biomaterials allowing for the recognition of their natural ligands with high performances. In addition, the powerful capability of biological chemical sensing systems could be preserved to some extent, which helps to enhance the performance of biosensors. Different types of functional biomaterials have been isolated from biological chemical sensing systems and successfully utilized as sensitive elements for the development of biosensors. For instance, olfactory sensory neurons and olfactory receptors have been isolated from animals or insects and have served as sensitive elements in biosensors for odor detection [26,48,49,50,51]. Similarly, taste bud cells and taste receptors have also been isolated from animals and applied in the biosensors for taste substance detection [27]. However, this approach has some limitations that hamper further development. The main problem is related to the purification of desired biomaterials, which have crucial influences in the specificity of the biosensors. It is usually time-consuming and expensive to achieve sufficient functional biomaterials for biosensors. In addition, it is also challenging to maintain their native function during the preparation and measurement process of biosensors. All these limitations make it difficult to develop a practical applicable or commercially available biosensors, especially for those in-field applications.

Fast advances in biotechnology provide an alternative approach for the preparation of functional biomaterials for biosensors. This approach can be used to achieve functional biomaterials by the expression of desired type of olfactory or taste receptors either in a heterologous cell system or a cell-free protein synthesis system. This allows for the preparation of functional biomaterials with desired types of olfactory or taste receptors. In addition, this approach makes it easy to graft tags in the prepared receptors, which could greatly facilitate the purification and immobilization of functional biomaterials to improve the performance of biosensors. For example, desired types of olfactory receptors have been expressed in human embryonic kidney (HEK) cells [52,53] and yeast [54,55,56,57] and utilized as sensitive elements for biosensors towards odorant detection. Taste receptors have also been expressed based on biotechnology to prepare functional biomaterials for the development of biosensors for taste substance detection [24,58,59,60]. However, this approach still suffered from the labor-intensive and complex purification process of functional biomaterials. In addition, the expression of receptors in a heterologous cell system usually led to cellular toxic effects that are mainly induced by the membrane incorporation and incompatibility of heterologous expressed olfactory or taste receptors. This results in low expression efficiency, which makes it difficult to improve the preparation efficiency of functional biomaterials. Therefore, cell-free protein synthesis is introduced as an alternative method to prepare functional biomaterials to address this limitation. The synthesis system provides all the necessary components for receptor synthesis such as amino acids, nucleotides, salts and energy-generating factors [61,62]. This cell-free system can not only avoid the cell toxic effect induced by receptor expression, but also could make the preparation process faster, which could mean that the whole expression process could finish within a few hours. Recently, olfactory and taste receptors (Figure 2) have been prepared by a cell-free protein synthesis method and coupled with different transducers for the development of biosensors towards chemical sensing [63,64]. This method could also help the right receptor protein folding via the modification of synthesis reaction conditions. However, it is still a big challenge to produce olfactory receptors in a highly efficient and convenient manner due to their hydrophobicity and dependence on a lipid bilayer environment [26].

### 2.2. Fabrication of Field-Effect Devices

Another key component of biosensors is the transducers. Appropriate transducers are also highly essential in order to convert the chemical signals sensed by the functional biomaterials into the measurable signals. For the development of biosensors using biomaterials for chemical sensing, mass-sensitive devices (e.g., quartz crystal microbalances, QCM, and surface acoustic wave, SAW) and field-effect devices (FEDs) are the most commonly used transducers [65,66,67]. Both of them can record the responsive signals from functional biomaterials upon exposure of chemical substances. Basically, FEDs function as transducers to detect the chemical signals sensed by functional biomaterials and transmit the responsive signals to the peripheral circuits for further signal processing [68]. Therefore, it is crucial to achieve very good and stable coupling between field-effect devices and functional biomaterials in order to develop biosensors with high performances. These biosensors are usually configured with corresponding measurement setup and peripheral circuits in order to readout, collect, and process the detected chemical signals. In this review, we will focus on the biosensors using FEDs as transducers, which mainly include field-effect transistor (FET), light-addressable potentiometric sensor (LAPS), and capacitive electrolyte–insulator–semiconductor (EIS) sensors (Figure 3).

Fast advances in the micro-fabrication process have greatly facilitated the design and fabrication of various specialized field-effect devices, which could be used as transducers for the development of biosensors towards chemical sensing. For example, FET can be fabricated via a standard micro-fabrication process on silicon wafer [20,69]. The mechanisms and structure of FET are schematically shown in Figure 3a. Usually, an insulator layer is first grown on the surface of silicon wafer via thermal oxidation, which can be used as the gate of FET devices. In some cases, the insulator layer was further deposited with a Si_3_N_4_ layer to improve the performance of FET devices. By the following, polyimide is often utilized to form a passivation layer in order to fix with a printed circuit board. Then, the source and drain electrodes are usually fabricated based on photolithography process. Finally, epoxy resin could be used to encapsulate FET devices, which is then fixed with a detection chamber allowing for the exposure of gate surface to the measurement solution inside the detection chamber. With this configuration, the chemical signals sensed by functional biomaterials can be coupled to the gate electrodes of FET, which are then transmitted to the peripheral circuit via the source and drain electrodes of FET.

The LAPS devices and EIS devices are also silicon-based FEDs, as shown in Figure 3b,c [70,71,72]. Both of them have the same structure of electrolyte–insulator–semiconductor. The difference between them is the measurement configuration. LAPS usually require a moveable focused light to realize addressable measurement on the desired points, while EIS do not require any light illumination during measurement. The structures of LAPS and EIS devices are much simpler than that of FET devices, which greatly facilitated the fabrication process. They are often fabricated based on silicon wafer, which is first thermal oxide with a layer of SiO_2_ on its surface to service as insulator layer. In most cases, the insulation layer surface was further grown with a layer of Ta_2_O_5_ or Si_3_N_4_ to improve their performance. Then, the oxide layer was removed from the rear side of the wafer, which is then deposited with a metal layer (e.g., Al or Au) to be utilized as Ohmic contact. Finally, the wafer was cut into separate small chips and fixed with a detection chamber. They can thus be applied to the development of biosensors by the immobilization of functional biomaterials onto the gate surface of FEDs exposed to the detection chamber.

### 2.3. Coupling of Functional Biomaterials with Field-Effect Devices

The coupling of functional biomaterials with FEDs has a significant influence on the performance of biosensors. It is thus highly essential to achieve highly efficient coupling between functional biomaterials and FEDs [23,73]. Highly efficient coupling means not only maintaining the structure and functions of functional biomaterials to make them suitable to serve as the sensitive elements for chemical sensing, but also to transduce the responsive signals into the output signals via FEDs. The output signals will then be further processed by the peripheral circuits [74,75]. Therefore, biosensors usually require the related peripheral circuits and measurement setup to realize the detection of chemical signals.

Functional biomaterials used for the development of biosensors are mainly divided into two categories, i.e., cellular/tissue biomaterials [28] and biomolecules [26,75]. As shown in Figure 4, for cellular/tissue biomaterials, it is ideal to provide a surface that is similar to the cell culture dish, which can provide good surface hydrophilicity and proper surface charges for cell or tissue culture and attachment. However, the surface of FEDs usually consists of silicon dioxide or metal oxide, which shows poor biocompatibility and makes it unsuitable for direct cell or tissue attachment and culture. To improve the biocompatibility of FEDs, a surface modification process is usually required before cell or tissue attachment as reported in some cases [28]. For example, poly-l-ornithine and laminin mixture with a proper rate have been utilized to treat the surface of FEDs to achieve better coupling between cells and FEDs [71]. However, at present, it is still a huge challenge to obtain ideal coupling between cell membrane and the surface of FEDs for the development of biosensors towards chemical sensing.

For biomolecules, highly efficient coupling with FEDs usually requires capturing functional biomolecules and avoiding the non-specific adsorption of unrelated molecules to improve the specificity of the biosensors. Current available immobilization approaches mainly include physical adsorption, covalent attachment via chemical reactions, and specific binding via couple molecular pairs, such as a biotin–avidin system. It is crucial to choose the optimal approach to develop biosensors according to the properties of functional biomaterials and surface characters of transducers, since each approach has its intrinsic advantages and disadvantages. For example, physical adsorption has the advantage of being simple, label-free, and reproducible, but it often suffers from the instability of coupling since it can be easily disrupted by minor changes in the microenvironment such as salt density. On the other hand, covalent attachment is much more stable and robust than physical adsorption. In addition, it provides an approach to regulate the surface density of biomolecules, which is very important for achieving optical performances of biosensors [63]. However, the process of covalent attachment is complex and usually require the modification of biomaterials or sensitive surface of transducers, which hamper their applications to some extent. Similarly, the biotin–avidin system can provide strong and robust noncovalent binding between biomolecules and the gate surface of FEDs, which shows very high affinity due to the specific strong interactions between avidin and streptavidin. The biotin–(strept)avidin complex is very strong and robust even in complex environments, which contribute greatly to the repeatability and reproducibility of biosensors. However, the biotin–avidin system also suffers from the complex labelling and reaction process. In general, to obtain the best performances of biosensors, the key point is to specifically couple the functional biomaterials with transducers with high specificity and high stability, which could help to avoid the nonspecific adsorption and generate stable and highly sensitive responsive signals. In addition, it is also very important to maintain the natural sensing functions of biomolecules, especially for those membrane receptors such as olfactory and taste receptors. A hydrophobic environment often needs to be provided, which is crucial to maintaining the chemical sensing function of membrane receptors [66].

## 3. Development of Field-Effect Sensors Using Biomaterials

Significant progress has been achieved in the field of field-effect sensors using biomaterials as sensitive elements and FEDs as transducers. There are three main types of FEDs that have been applied in the development of biosensors using biomaterials for chemical sensing, which include FET, LAPS, and EIS sensors. Each of them has shown promising prospects in various applications.

### 3.1. FET-Based Biosensors Using Biomaterials

The most commonly used FEDs is FET, in which the gate surface can be modified with various charge-sensitive layers for the sake of detecting charged biomolecules as well as potential changes induced by excitable cells such as neurons [76]. The obvious advantages of FET come from its innate signal amplification capability, which has shown promising prospects for the detection of weak biological electrical signals. The earliest study of applying FET to biosensors using biomaterials was reported in 2000, in which FET was utilized to couple with the antenna of Colorado potato beetles to form a bioelectronic interface for the detection of a volatile marker (i.e., (Z)-3-hexen-1-ol) of plant damage, as shown in Figure 5 [20]. This biosensor was able to detect the beetle-damaged plants with high performance in the field, which represents a powerful tool for plant protection and food safety. FET can provide a particular reliable joining between an insect antenna and transducer, which makes it ideal for recording the responsive electrical signals from antennae of beetle in response to (Z)-3-hexen-1-ol [77,78]. In short, the potential changes in insect antenna induced by the exposure of specific volatile compounds were recorded by monitoring the changes in the drain current from the FET source and drain electrodes, which are dependent on the concentration of specific chemical volatile compounds. In addition, the small size of biosensors based on FET devices makes it possible to develop portable instruments for the in-field applications such as the detection of explosive compounds in the field of public safety, plant damage detection in the field of plant protection, and smoke detection for building safety.

At the molecule level, single wall carbon nanotube (swCNT) has been used to modify the gate surface of FET to generate swCNT-FET, which has been used as transducer and combined with DNA molecules to develop biosensors for chemical sensing [79]. It is indicated that this biosensor with hybrid nanostructure is capable of detecting specific volatile compounds with high sensitivity and specificity. It has been proven that distinct responsive signals can be recorded from swCNT-FET coupled with different bases of DNA molecules. The DNA base sequence-dependent responses suggested that this biosensor is suitable to be utilized to construct gas sensor array towards electronic noses since the responsive signals to the compounds are mainly dependent on the specific base sequences of ssDNA molecules. Similarly, olfactory receptor protein (i.e., hOR2AG1) has been immobilized onto the gate surface of FET that had been previously modified with swCNT [52] or carboxylated-polypyrrole nanotubes (CPNT) [80] for the development of biosensors in order to detect specific odorants with high sensitivity. Similarly, taste receptors have also been attached onto the gate surface of swCNT-FET or CPNT-FET to develop biosensors for the detection of bitter compounds [24,59]. The basic mechanism of these biosensors using receptors rely on the specific interactions between receptors and their ligands, which can often be measured by monitoring the changes in the drain-source current of FETs (Figure 6). These biosensors using human taste receptor protein for bitter compound detection have been applied for the detection of real food samples, which provide valuable tool and show promising prospects in the field of food safety.

### 3.2. LAPS-Based Biosensors Using Biomaterials

LAPS is a surface potential detector, which is suitable for use as a transducer for the development of cell-based biosensors [71,81,82]. It is able to record the changes in extracellular potentials of chemical sensitive cells such as olfactory sensory neurons and taste receptor cells. LAPS has the advantage of having a flat surface and light addressability, which make it ideal for cell measurement. Cells can be cultured randomly on the LAPS surface and a focused light is used to choose the desirable cells for measurement. This overcomes the limitations of FETs, which usually require cells to be cultured precisely on the gate area of the devices.

LAPS has been used to develop various biosensors by the combination with different types of chemical sensitive cells. For instance, olfactory sensory neurons isolated from rat epithelium have been cultured on the LAPS surface to develop a biosensor towards the detection of odorants or neurotransmitters, such as acetic acid and glutamic acid [50,51]. In addition, LAPS has also been reported to be able to record the responsive signals from an intact rat olfactory epithelium induced by different odorants [83,84]. However, the utilization of biomaterials directly originating from animals usually limited by their intrinsic properties such as unknown types of olfactory receptors existing in the biomaterials. To address this issue, bioengineered olfactory receptor neurons expressed with well-defined olfactory receptors were employed to serve as sensitive elements for biosensors towards odorant detection [85]. It is reported that this biosensor based on bioengineered olfactory receptor neurons can be used to detect the specific target odorant in a dose-dependent manner. Furthermore, HEK-293 cells expressed with a specific olfactory receptor, ODR-10, were utilized to couple with LAPS to develop a biosensor for the detection of specific odorant, diacetyl. It has been proven that the measurement of the cell acidification signals recorded by LAPS from single cells can also be used as the responsive signals for the detection of specific odorant stimulation [60].

Similarly, taste cells isolated from rat tongue have been cultured on the LAPS surface to develop biosensors for various taste signals such as bitter [86] and acid [87]. For example, the LAPS surface has been modified with a thin serotonin-sensitive polyvinyl chloride (PVC) membrane, which has been applied in the research of taste cell-to-cell communications via the monitoring of serotonin released from single taste cells [88]. It is reported that this biosensor was able to record the cell membrane potential changes as well as serotonin release from single taste cells in response to acid stimulation and taste mixture (bitter and sweet). In addition, the LAPS surface was modified with a layer of ATP-sensitive aptamers and applied in the detection of ATP release as well as membrane potential changes from single taste bud cells under taste mixture stimulations (Figure 7) [89,90]. It has been proven that this biosensor was able to detect local ATP secretion from a single taste cells in a dose-dependent manner. Biosensors based on LAPS provide a novel and powerful approach to researching taste sensation, which could potentially contribute to the understanding of taste signal transduction mechanisms and cell-to-cell communication.

### 3.3. EIS-Based Biosensors Using Biomaterials

Similar to LAPS, capacitive EIS sensors belong to the FEDs category and are a kind of charge-sensitive devices. EIS sensors are able to detect surface charge changes induced by the attachment of charged molecules onto the sensor surface. The most common applications of EIS sensors are related to the label-free detection of pH changes [91,92], ion concentrations [93,94,95,96,97,98], charged molecules [99], and charged nanoparticles [100,101]. In principle, the attachment of charged molecules or the binding of receptor and ligand occurring on the gate surface of an EIS sensor will lead to the redistribution of surface charge, which will, in turn, result in changes in the space–charge distribution in the semiconductor layer of the EIS sensors. These changes can be reflected by the changes in the output signals of the ES sensors. The decisive advantages of EIS sensors are their simple structure and low cost, which can be fabricated in a convenient and low-cost manner due to the unnecessary involvement of photolithographic process steps or complicated encapsulation procedures. In addition, the capability of surface charge detection makes them suitable for use as transducers for the development of biosensors towards label-free chemical sensing. For instance, EIS sensors have been combined with an olfactory receptor, ODR-10, to develop a biosensor for the detection of a specific odorant, diacetyl [63]. The mechanism of this biosensor was schematically shown in Figure 8a. To improve the coupling efficiency of olfactory receptors with the EIS sensor, the olfactory receptors were prepared using a cell-free protein expression system and fused with a His_6_-tag to realize the on-chip purification of sensitive elements based on EIS sensor modified with anti-His_6_-tag aptamers. The responsive signals induced by the specific binding between olfactory receptor and its ligand were measured by the monitoring the capacitance changes in the EIS sensor, which is performed by the capacitance − voltage (*C* − *V*) and constant-capacitance (ConCap) measurements. It has been proven that this biosensor is able to detect diacetyl in a linear concentration-dependent manner at concentrations ranging from 0.01 nM to 1 nM with a detection limit of 0.01 nM (Figure 8b). This biosensor has great potential to be applied in various fields related to chemical sensing such as biomedicine, food safety, and environmental protection.

## 4. Conclusions and Prospects

With fast advancement in the microfabrication process, more and more FEDs have been designed and fabricated for various applications. The increasing utilization of FEDs as transducers and functional biomaterials as sensitive elements as part of the development of biosensors for chemical sensing has become a recent trend, which is attracting more and more attention. Biosensors based on FEDs have also shown promising prospects and potential applications in a wide range of fields such as biomedicine, food safety, and environmental protection. However, there are also some limitations that hamper the further development and applications of field-effect sensors using biomaterials. At present, the challenges faced regarding the further development of field-effect sensors using biomaterials mainly include: (1) how to obtain sufficient functional biomaterials that are suitable to serve as sensitive elements, (2) how to fabricate microscale/nanoscale FEDs with sizes that are comparable to the sizes of functional biomaterials, and (3) how to improve the coupling efficiency of biomaterials and transducers as well as the responsive signal transduction efficiency. In the near future, the development of biosensors based on FEDs will probably part of the method of addressing the challenges mentioned above.

Field-effect sensors using biomaterials have shown a powerful capability for chemical sensing, and can not only be used as a novel approach to chemical sensing, but can also be applied in the research for mechanisms of chemical sensations. The final goal of future research and development on field-effect sensors using biomaterials is to improve their performances for chemical sensing in complex environments, which includes improvements on sensitivity, specificity, repeatability, and stability. This usually requires the incorporation of multiple technique advancements in different fields such as biotechnology, nanotechnology, and microfabrication processes. For instance, progress in nanotechnology and microfabrication processes allows for the micro/nano FEDs that could facilitate the coupling with biomaterials. Similarly, advancement in biotechnology could provide novel approaches for the preparation of functional biomaterials that are more suitable to being used as sensitive elements in the development of field-effect sensors using biomaterials for chemical sensing. It is expected that these advancements will greatly contribute to the further development and applications of field-effect sensors using biomaterials.

## Figures and Tables

**Figure 1 sensors-21-07874-f001:**
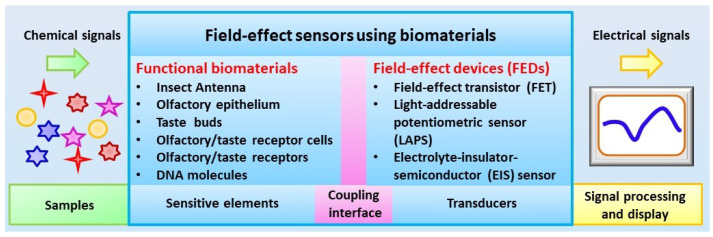
Schematic diagram of configurations of field-effect sensors using biomaterials.

**Figure 2 sensors-21-07874-f002:**
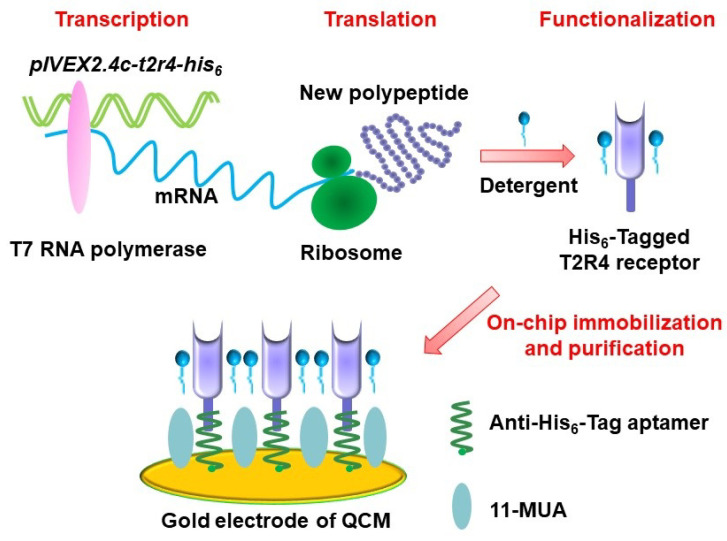
Schematics of preparation of a bitter receptor from cell-free protein expression system for chemical sensing. (Reprinted with permission from ref. [64]. Copyright 2020 Elsevier).

**Figure 3 sensors-21-07874-f003:**
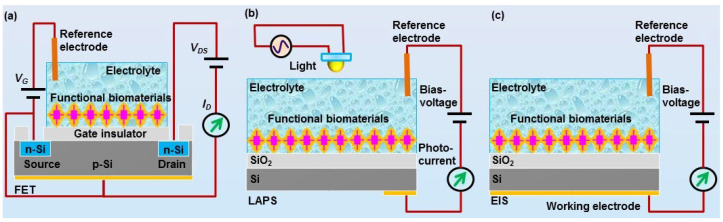
Schematics of field-effect devices utilkized for the development of field-effect sensors using biomaterials for chemical sensing, including (**a**) field-effect transistor (FET), (**b**) light-addressable potentiometric sensor (LAPS), and (**c**) electrolyte-insulator-semiconductor (EIS) sensor.

**Figure 4 sensors-21-07874-f004:**
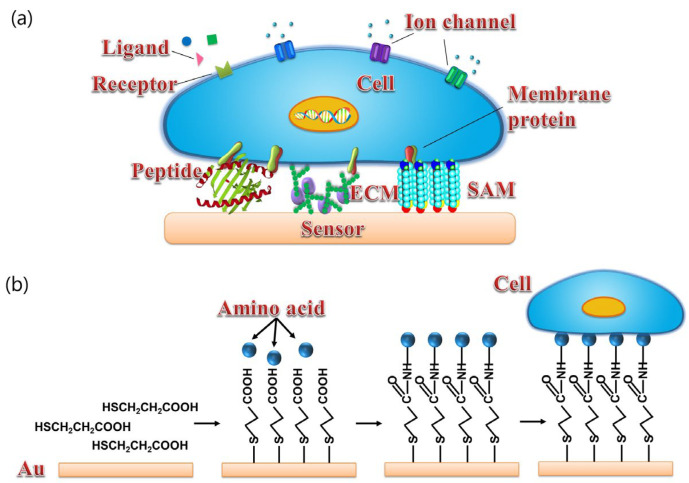
(**a**) Schematics of different surface modification of transducers for cell coupling with sensor including peptide, ECM, and SAM. (**b**) Schematic diagram of cells coupled with gold surface via SAM. (Reprinted with permission from ref. [28]. Copyright 2014 American Chemical Society).

**Figure 5 sensors-21-07874-f005:**
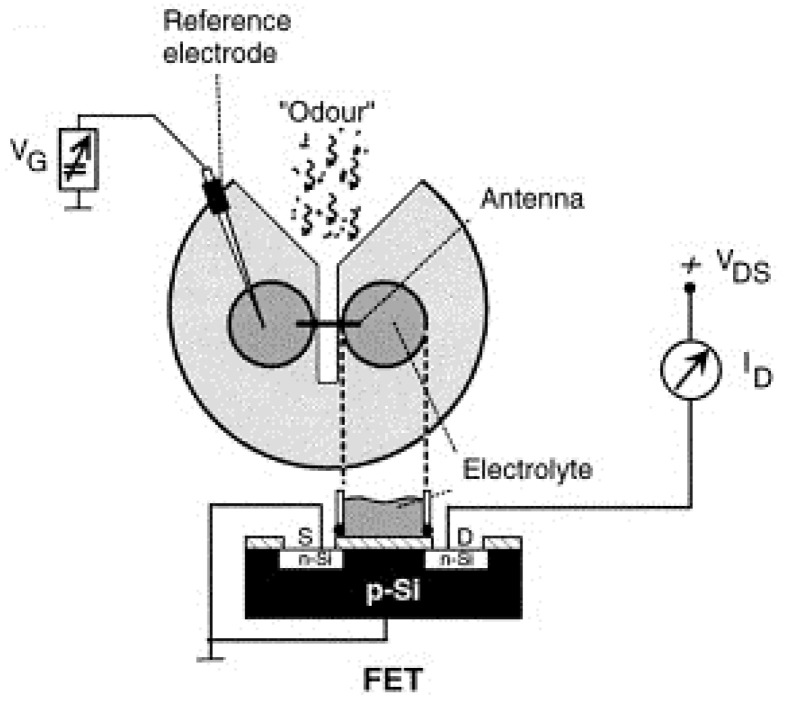
Schematics of field-effect sensors using insect antenna as sensitive element and FET as transducer for volatile marker detection towards plant protection. (Reprinted with permission from ref. [20]. Copyright 2000 Elsevier).

**Figure 6 sensors-21-07874-f006:**
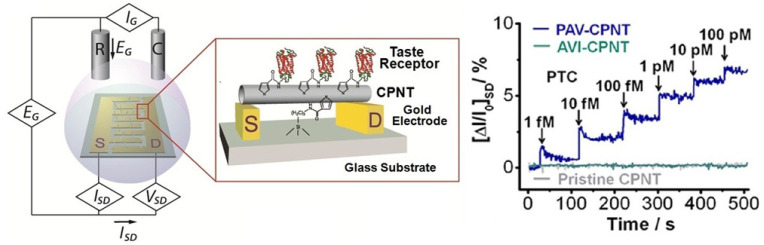
Schematic diagram of a field-effect sensor using taste receptors and its responses to different concentrations of bitter substances. (Reprinted with permission from ref. [59]. Copyright 2013 American Chemical Society).

**Figure 7 sensors-21-07874-f007:**
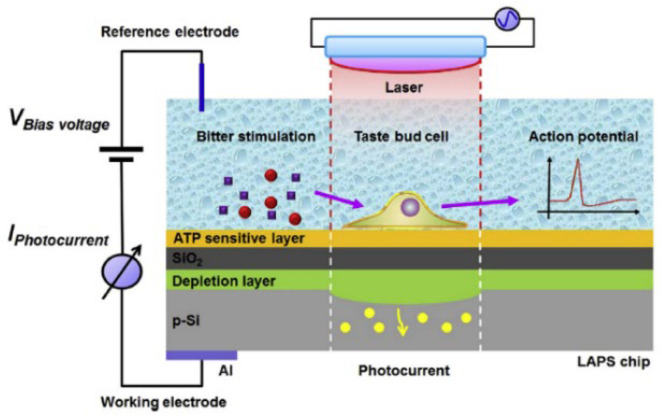
Schematics of a field-effect sensors using LAPS as transducer for the detection of ATP release and membrane potential changes from single taste bud cell. (Reprinted with permission from ref. [90]. Copyright 2018 Elsevier).

**Figure 8 sensors-21-07874-f008:**
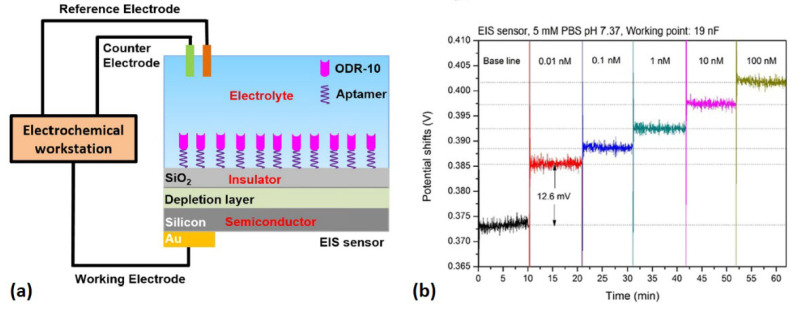
(**a**) Schematics of an EIS sensor using olfactory receptors for the detection of specific odorant. (**b**) Responses of this EIS sensor to different concentrations of odorant. (Reprinted with permission from ref. [63]. Copyright 2019 Elsevier).

## Data Availability

Not applicable.

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
