# Peer review of "Field-Effect Sensors Using Biomaterials for Chemical Sensing"

_sensors, 2021, doi:10.3390/s21237874_

Round 1

Reviewer 1 Report

In this mini-review, the authors summarised some development of field-effect sensors for chemical detection. The ms reads very well.

If some of the presented works relate to biomimetic devices, many results correspond however to devices where biological materials are directly used. They correspond more to “bio-utilisation”, nameyly utilisation of biological materials, rather than “biomimetism”, “biomimetics” or “biomimicry”. I must admit that I think the authors overused the term “biomimetic”. I recommend the authors to use much less “biomimetic” in their ms. Actually, “biomimetic” should be used where natural systems were copied but not directly used.

One example is the DNA swCNT sensors presented in lines 300-302. This device is adorn with DNA but is not a mimic nor inspired by any natural/biological systems/organisms and specifically not the olfactory and taste cells and receptors. Though in lines 306-307, the authors used the qualificative “biomimetic” for these sensors. A bit further, taste receptors were attached onto the gate of the swCNT FET or CPNT-FET. This is not a biomimetic sensor. This sensor relies on the utilisation of biological material, which is different.

Similarly, the LAPS-based sensors described in lines 322-329 are not biomimetic. The authors measured the response of the actual biological materials. Same problem with the devices described in lines 330-338, the ones in lines 347-361 and others studies mentioned in this review.

From a story telling point of view, the authors should either focus only on the devices that are mimic of nature (olfactory and taste cells) or enlarge the scope of their review.

In case the authors decide to enlarge the scope of the review, I suggest the title to be changed to “Biological and biomimetics Field-Effect Sensors for Chemical Sensings” or anything equivalent that would reflect my comment.

Out of 73 references, I counted up to 24 self-citations. The authors should probably decrease this ratio. In addition, I pointed out some paragraphs where references should be added (see next paragraph and specific remarks).

Many chemical sensors based on metaloxyde semiconductor (MOS) were developed based on natural porous structures, through bioinspiration (see Dong et al., Fabrication and gas sensitivity of SnO2 hierarchical films with interwoven tubular conformation by a biotemplate-directed sol-gel technique, Nanotechnology 17, 2006; Zhu et al., A simple and effective approach towards biomimetic replication of photonic structures from butterfly wings, Nanotechnology 20, 2009; Song et al., Fabrication and good ethanol sensing of biomorphic SnO2 with architecture hierarchy of butterfly wings, Nanotechnology 20, 2009; Song et al., Controllable synthesis and gas response of biomorphic SnO2 with architecture hierarchy of butterfly wings, Sensors and Actuators B 145, 2010; Tian et al., A highly sensitive room temperature H2S gas sensor based on SnO2 multi-tube arrays bio-templated from insect bristles, Dalton Transactions 44, 2015). Due to the close relationship between the focus of this mini-review and these studies, the authors could consider mentioning such research briefly in their introduction. More generally, other types of biomimetic chemical sensors could have been referenced in the paper (such as Potyrailo et al., Towards outperforming conventional sensor arrays with fabricated individual photonic vapour sensors inspired by Morpho butterflies, Nature Communications 6, 2015; Poncelete et al., Vapour sensitivity of an ALD hierarchical photonic structure inspired by Morpho, Bioinspiration & Biomimetics 11, 2016; Rasson et al., Vapor sensing using a bio-inspired porous silicon photonic crystal, Materials Today Proceedings 4, 2017). This may help addressing the issues related to the scope and the self-citation mentioned here above.

There are several language oddities in the abstract (e.g., “for the first” L20 or “by the following” L22) and the paper (e.g., “surround environment” L38-39, “origination” L45, “costive” L129, “contribute great” 259 or “the decisive advantages of FET is originated” L277-278). I am no native speaker. Maybe the authors are right but they may want to check these.

Specific remarks:

L40: “enemies” should probably be replaced by “predators”.

L60-63: references may be needed here.

L131: I believe “challenge” should be replaced by “challenging”.

L179: “focusing” should be replaced by “focus”.

L180: “includes” should bereplaced by “include”.

L219-226: some references should be included here to support these statements.

L291: “make” should be substitute by “makes”.

L359-360: a “to” is probably missing between “contribute” and “the”.

L371: a space is missing between “concentrations” and “[65-70] and another one between “nanoparticles” and “[72,73]”.

L389: wouldn’t a hyphen be missing between “constant” and “capacitance”?

Reviewer 2 Report

In this work Wu and coworkers summarize recent advances in the development of biomimetic field-effect sensors for chemical sensing,
 emphasizing the use of functional biomaterials as sensing elements including olfactory/taste cells and receptors. 

the authors intruduce principles and approaches exploited for developing biomimetic field-effect sensors, then present different types of biomimetic field-effect sensors, such as field-effect transistor, light-addressable potentiometric sensor and capacitive electrolyte–insulator–semiconductor sensors, and finally discuss current limitations, challenges and future trends of biomimetic field-effect sensors for chemical sensing.

the work is well written and clear and the storytelling is linear and convincing. 
overall the manuscript provides an exaustive review on the specific topic, combining newly developed schematics and figures to explain the main conpects together with figure panels taken from published works.

the authors should expand a bit the presentation of the main results reported in the literature by includind a few more figures assembled starting from references cited in the manuscript, 
for what concerns the following sections:

2.1. Preparation of functional biomaterials
2.3. Coupling of functional biomaterials with field-effect devices
3.1. FET-based biomimetic sensors

basically adding at least one more figure for each section

provided these improvements are implemented in the manuscript, I fell the work could be suitable of publication in Sensors 

Round 2

Reviewer 1 Report

The authors have replied to all my comments on a very satisfactory way.

I would like to congratulate them for their work and manuscript.

Reviewer 2 Report

Revisions implemented paper abitabile for pubblicato. In sensors